# Intensity of Depression Symptoms Is Negatively Associated with Catalase Activity in Master Athletes

**DOI:** 10.3390/ijerph20054397

**Published:** 2023-03-01

**Authors:** Larissa Alves Maciel, Patrício Lopes de Araújo Leite, Patrick Anderson Santos, Lucas Pinheiro Barbosa, Sara Duarte Gutierrez, Lysleine Alves Deus, Márcia Cristiane Araújo, Samuel da Silva Aguiar, Thiago Santos Rosa, John E. Lewis, Herbert Gustavo Simões

**Affiliations:** 1Postgraduate Program in Physical Education, Catholic University of Brasília, Brasília 71966-700, Brazil; 2Postgraduate Program in Physical Education, Federal University of Mato Grosso-UFMT, Cuiabá 78060-900, Brazil; 3Department of Psychiatry and Behavioral Sciences, Miller School of Medicine, University of Miami, Miami, FL 33316, USA

**Keywords:** depression, master athletes, redox balance, catalase

## Abstract

Background: This study examined associations between scores of depression (DEPs), thiobarbituric acid-reactive substances (TBARS), superoxide dismutase (SOD), and catalase activity (CAT) in master athletes and untrained controls. Methods: Participants were master sprinters (MS, *n* = 24; 50.31 ± 6.34 year), endurance runners (ER, *n* = 11; 51.35 ± 9.12 year), untrained middle-aged (CO, *n* = 13; 47.21 ± 8.61 year), and young untrained (YU, *n* = 15; 23.70 ± 4.02 year). CAT, SOD, and TBARS were measured in plasma using commercial kits. DEPs were measured by the Beck Depression Inventory-II. An ANOVA, Kruskal-Wallis, Pearson’s, and Spearman’s correlations were applied, with a significance level of *p* ≤ 0.05. Results: The CATs of MS and YU [760.4 U·μL 1 ± 170.1 U·μL 1 and 729.9 U·μL 1 ± 186.9 U·μL 1] were higher than CO and ER. The SOD levels in the YU and ER [84.20 U·mL^−1^ ± 8.52 U·mL^−1^ and 78.24 U·mL^−1^ ± 6.59 U·mL^−1^ (*p* < 0.0001)] were higher than CO and MS. The TBARS in CO [11.97 nmol·L^−1^ ± 2.35 nmol·L^−1^ (*p* < 0.0001)] was higher than in YU, MS and ER. MS had lower DEPs compared to the YU [3.60 ± 3.66 vs. 12.27 ± 9.27 (*p* = 0.0002)]. A negative correlation was found between CAT and DEPs for master athletes [r = −0.3921 (*p* = 0.0240)] and a weak correlation [r = −0.3694 (*p* = 0.0344)] was found between DEPs and the CAT/TBARS ratio. Conclusions: In conclusion, the training model of master sprinters may be an effective strategy for increasing CAT and reducing DEPs.

## 1. Introduction

Master athletes are those over 35 years old who maintain a training routine, compete in national and/or international sports championships, and represent a distinct portion of the middle-aged and elderly population [1]. These master athletes undergo intense physical training routines (from 3 to 6 sessions per week, totaling approximately 10 h or more of weekly training). The training programs of elite masters sprinters, for example, are mainly characterized by high-intensity, low-volume sessions based mainly on anaerobic pathways [1,2]. Additionally, the weekly training of such sprinters usually includes two sessions for speed (i.e., short sprints and long sprints for speed endurance), strength (i.e., weight lifting), power (i.e., plyometric exercises), and stretching and flexibility. A few low-volume, moderate-intensity cardiovascular sessions can also be performed between high-intensity anaerobic training sessions. Otherwise, the training programs of elite master endurance athletes are based mainly on low-intensity, high-volume sessions, relying primarily on aerobic pathways. Exercise modes and workloads are selected individually depending on the athlete’s training season [1,2]. Several studies have shown that master athletes have better physical performance, body composition, lipid profile, blood glucose control, and attenuated biological aging when compared to untrained peers, and have thus been referred to as a healthy aging model [3,4].

The healthy aging process of master athletes may be related to several biochemical mechanisms, including lower levels of oxidative stress and increased antioxidant defense [5]. A decreased antioxidant defense, on the other hand, has been identified as a part of the pathogenesis of depression, a multifactorial disease linked to the aging process [6]. In situations in which the antioxidant system is impaired, reactive oxygen species can damage lipids, proteins, and DNA. In addition, pro-inflammatory cytokines (IL-6, TNF-ӱ) increase the activity of indoleamine 2,3-dioxygenase (IDO), an enzyme involved in the synthesis of kynurenine from tryptophan. Kynurenine in turn appears to have potential neurotoxic action, since kynurenine is transformed into 3-monooxygenase (KMO), forming kynurenine into 3-hydroxykynurenine and 3-hydroxyanthranilic, precursors of quinolinic acid. This acid is considered a metabolite that leads to excitotoxicity for the central nervous system and induces oxidative stress. Thus, some studies have shown that catalase seems to block the toxicity generated by 3-hydroxykynurenine [7,8,9].

Antioxidant defense, in turn, protects cells by removing free radicals. This antioxidant system comprises different types of functional components, such as superoxide dismutase (SOD) and catalase (CAT) [10]. SOD acts as a primary cellular defense against free radicals since it catalyzes the reduction of SO to oxygen and hydrogen peroxide. CAT is an antioxidant enzyme present in almost all aerobic organisms. Its function is to break two molecules of hydrogen peroxide into one molecule of oxygen and two molecules of water [10,11]. In our previous studies, we have shown that master athletes have greater catalase activity than their non-athlete peers, in addition to other antioxidant enzymes such as superoxide dismutase (SOD) [3,4].

However, findings on catalase and its relationship with the intensity of depression symptoms (DEPs) are still inconsistent. According to Tsai and Huang [12], catalase activity is increased in patients in the acute phase of depression. On the other hand, in a meta-analysis by Jimenez-Fernandez [13], the differences in catalase levels among depressed and non-depressed people were not significant.

Furthermore, the substance reactive to thiobarbituric acid (TBARS) is an enzyme that is abundant in the depressive process. TBARS is the main method to quantify the end products of lipid peroxidation, being considered a pro-oxidant enzyme used to measure the oxidative stress of tissues and cells [14]. This oxidative stress is defined as an imbalance between pro- and antioxidant molecules.

To the best of our knowledge, no research has been conducted on the relationship between catalase activity, oxidative stress, and the intensity of depression symptoms in people who have followed a training regimen their entire lives, such as master runner athletes. Therefore, we aimed to analyze catalase, oxidative stress, and the intensity of depression symptoms in master athletes, their non-athlete peers, and a young control group. We hypothesized that master athletes have higher catalase activity and lower intensity of depression symptoms when compared to their non-athlete peers and the youth control group. It is also hypothesized that there is a negative correlation between catalase activity and the intensity of depression symptoms.

## 2. Materials and Methods

### 2.1. Ethical Approval

The study was approved by the Human Research Ethics Committee. All procedures were carried out according to the principles of the Declaration of Helsinki (466/2012). All subjects who agreed to participate in the study provided written informed consent, which had been clearly explained before participation.

### 2.2. Participants

The total sample (*n* = 63) was composed of 35 elite male master athletes at regional, national, and international levels and 28 untrained individuals. The master athletes were subdivided into master sprint athletes (MS, *n* = 24) from the 100 m, 200 m, 400 m, and 110 m hurdles, among others, and endurance runners (ER, *n* = 11) from 5 km to marathons and triathletes. The control groups consisted of young untrained (UY, *n* = 15) and middle-aged untrained controls (CO, *n* = 13). The youth sample was entirely collected in Brazil. These were mostly single college students. Master athletes were recruited from participants in the Brazilian Master Athletics Championship (São Bernardo do Campo, Brazil, 2018), Grandprix Del Mercosur (Montevideo, Uruguay, 2019), and World Master Indoor Athletics Championship (Torún, Poland, 2019). The inclusion criteria for master athletes were: (1) systematic training for at least 10 years; and (2) active participation in national and/or international competitions until the date of data collection. The non-athlete subjects of the control group (young and middle-aged) were recruited through pamphlets and electronic advertisements in the city of Brasília-DF, Brazil, and met the inclusion criteria of not being trained and being healthy. The exclusion criteria for all participants were: (1) a history of cardiometabolic diseases; (2) a history of inflammatory disease and cancer; (3) a smoker; and (4) regular drug use, including hormone replacement therapy.

### 2.3. General Procedures

Data were collected in the laboratory between 7 and 9 am, and all volunteers had not exercised in the previous 12 h and had fasted for at least 8 h. The collection protocol consisted of (a) anamnesis, to collect data referring to the health history and history of training and/or physical activity; and (b) an assessment of the intensity of depression symptoms, for which data were collected using the Beck Depression Inventory-II (BDI-II). The instrument has 21 items, and for each of them, there are four response statements, among which the subject chooses the most applicable to describe how she has been feeling in the last two weeks, including the test date [15]. These items refer to levels of intensity of depression symptoms, and the total score is the result of the sum of the individual items, reaching a maximum of 63 points. The final score is classified into minimal, mild, moderate, and severe levels, thus indicating the intensity of depression. The questionnaires were administered one day before the competitions at the athletes′ accommodations, which were usually close to the competition venue. The same researcher performed all of these assessments; and (c) collection of venous blood from the antecubital vein using a 4 mL vacutainer (with EDTA), with blood gradient centrifugation (Sirius 4000, Sieger, Brazil) for 15 min at 3800 rpm for plasma and serum isolation, and storage in a freezer (−80 °C) for further plasma analysis of catalase, superoxide dismutase (SOD), and TBARS.

### 2.4. Antioxidant Parameters

The three antioxidant parameters used in this study were measured using commercial kits and following the manufacturer′s protocol. The SOD activity was measured using the SOD assay kit (Sigma Aldrich^®^, California, USA), with a final spectrophotometric reading at 450 nm; the CAT activity was measured using the Amplex TM Red Catalase assay kit (Thermofisher Scientific^®^, California, USA), with a final spectrophotometric reading after one minute of incubation at 560 nm.

### 2.5. Lipid Peroxidation (TBARS)

The protocol used in the present study is adapted from Ohkawa et al. (1979). Briefly, serum samples were diluted in 320 μL MiliQ H_2_O (1:5) and added 1 mL of trichloroacetic acid (TCA) 17.5%, pH 2.0, following the addition of 1 mL of thiobarbituric acid (TBA) 0.6%, pH 2.0. After homogenization, the samples were kept in a water bath for 30 min at 95 °C. The reaction was interrupted with the immersion of the microtubes in ice and the addition of 1 mL of TCA 70%, pH 2.0, and another incubation for 20 min at room temperature. After centrifugation (3000 rpm for 15 min) the supernatant was removed to new microtubes and taken to spectrophotometry reading at 540 nm. The concentration of lipid peroxidation products was calculated using the molar extinction coefficient equivalent for malondialdehyde (MDA − equivalent = 1.56 × 10 5 M − 1 cm − 1).

### 2.6. Statistical Analysis

The data were analyzed for normality and homogeneity using the Shapiro-Wilk test and the Levene test, respectively. The data were expressed as mean, standard deviation (±), minimum, 25% percentile, median, 75% percentile, and maximum. A one-way ANOVA followed by Tukeys post hoc was applied for comparisons among studied groups for age, catalase and Tbars variables. Kruskal-Wallis with Dunn’s test of multiple comparisons was applied to compare the groups on depression and SOD variables. The Spearman coefficient correlation was used to verify the association between catalase activity and the intensity of depression symptoms. The significance level was set at 5% (*p* ˂ 0.05), and all procedures were performed using GraphPad Prism (v7.0, California, USA). In order to assess the clinical importance of results, the effect size was calculated and classified either as small (r = 0.2 to 0.49), moderate (r = 0.5 to 0.79) or large (r ˃ 0.8) [15]. The sample size for a priori statistical power of 80% (1 − β = 0.80) indicated 20 participants for a significance level of 5% (α = 0.05) and small effect size (f = 0.4). Thus, we chose a sample of 80 subjects (20 for each studied group) [16].

## 3. Results

The characterization of the sample, the intensity of depression symptoms, the CAT, the SOD level, and the TBARS are expressed in Table 1 as mean and standard deviation. The intensity of depression symptoms in the YU (12.27 ± 9.27) was higher than in the MS group (3.60 ± 3.66; *p* = 0.0002) and CO (4.61 ± 2.56; *p* = 0.002). The CAT of MS and YU [760.4U · μL 1 ± 170.1 U·μL^−1^ and 729.9 U · μL 1 ± 186.9 U · μL 1] were higher than CO and ER [410.3 U · μL 1 ± 67.24 U · μL 1 and 528.8 U · μL 1 ± 103.2 U · μL^−1^ (*p* < 0.0001)]. The SOD level in the YU and ER was higher than CO and MS [84.20 U·mL^−1^ ± 8.52 U·mL^−1^ and 78.24 U·mL^−1^ ± 6.59 U·mL^−1^ (*p* < 0.0001)]. The TBARS in CO [11.97 nmol·L^−1^ ± 2.35 nmol·L^−1^ (*p* < 0.0001)] was higher than in YU, MS, and ER (Table 1).

Furthermore, a negative correlation was found between CAT and the intensity of depression symptoms for the entire group of master athletes [r = −0.3921 (*p* = 0.0240)] (Figure 1).

On the other hand, the CAT/TBARS ratio has a negative correlation with symptoms of depression [r = −0.3694 (*p* = 0.0344)] (Figure 2).

The SOD/TBARS ratio was not correlated with the intensity of depression symptoms [r = 0.1439 (*p* = 0.4319)] (Figure 3).

The relationships between the intensity of depression symptoms and SOD [r = 0.3320 (*p* = 0.06)] and TBARS [r = 0.0900 (*p* = 0.61)] were not statistically significant.

## 4. Discussion

This was the first study to assess the intensity of depression symptoms and their relationship with TBARS, SOD, and CAT activity in master athletes. Our main findings were that: (i) the young control group presented greater intensity of depression symptoms in comparison with both the master athletes from sprints and the middle-aged untrained control group; (ii) the young control group did not differ from the endurance runners in terms of intensity of depression symptoms; (iii) CAT activity was negatively associated with the intensity of depression symptoms in master athletes; (iv) the CAT/TBARS ratio was a negative correlation with symptoms of depression.

There is an increase in the prevalence of depression in young adults (18 to 25 years old), especially during the college period. One of the possible explanations for the phenomenon is that college students, when seeking academic performance, start to neglect their time, their social relationships, and their well-being, and, as a consequence, they also reduce their levels of physical activity [17,18,19,20]. All these changes can generate instability that can, therefore, contribute to the reduction of social support and increase in stress, which are known to contribute to the emergence of mental disorders [21].

On the other hand, while depression is among the most prevalent age-related mental conditions, the literature places the master athlete as a model of healthy aging, as long as he has a balanced lifestyle with healthy eating, stress control, and regular exercise for many years [3,4,22,23]. In this regard, our findings revealed that master sprint athletes have lower levels of intensity depression than untrained young people. Previously, it was evidenced in a meta-analysis that high-intensity neuromuscular training is more effective in reducing the intensity of depression symptoms when compared to aerobic exercise [24]. It is important to note that our master sprint athletes require more intense neuromuscular solicitations than our endurance athletes.

In this regard, neuromuscular/resistance training would increase the release of brain-derived neurotrophic factor (BDNF) from muscle contraction, reaching the brain and activating multiple signaling pathways, starting to regulate the expression of antioxidant molecules [25,26]. In addition, BDNF participates in the pathophysiological mechanism of depression. Since there is signaling for an increase in NF-kB, this would increase oxidative stress, causing an increase in pro-inflammatory cytokines (IL-1 and IL-6) and a decrease in BDNF, resulting in a decrease in brain cell neurogenesis [27].

Furthermore, Schuch et al. (2014) demonstrated the effects of 3 weeks of physical exercise in severely depressed hospitalized patients; those who had a decrease in TBARS levels after the exercise protocol was applied [28]. This result is in line with the present, which demonstrated lower TBARS levels in master sprint and endurance athletes when compared to the middle-aged group, confirming a possible adjuvant antioxidant effect in combating the intensity of depressive symptoms in this population.

However, the findings on the activity of antioxidant enzymes and depression are controversial. Increased activities have been detected in some studies, but on the other hand, several studies have published mixed or negative results for catalase activity in depression compared to healthy control groups [12,26,29].

Catalase (CAT) is an enzyme that catalyzes the breakdown of hydrogen peroxide into water and oxygen, mediating signaling in cell proliferation, apoptosis, carbohydrate metabolism, and platelet activation [30]. Humans with low catalase levels are at increased risk for diabetes and altered lipid and carbohydrate metabolism [31]. Some studies that have examined catalase activity in depressed patients have found increased levels of catalase activity during acute episodes of depression compared to healthy volunteers [31]. Szuster-Ciesielska et al. [32] also detected increased serum catalase activity in patients with major depression. The increase in catalase activity may reflect a compensatory mechanism since, during depressive disorders, there is an increase in oxidative and nitrosative stress (O&NS) pathways. The catalase would be increased to attenuate the induced O&NS pathways and is congruent with the role of oxidative free radical signaling [32].

On the other hand, some clinical studies reported a decrease in catalase activity during depressive episodes [31]. According to a study by Bhatt et al., mild and chronic stress led to decreased levels of catalase in the brain tissues of stressed mice; however, treatment with antidepressants had beneficial effects and increased catalase levels in these mice [31]. Additionally, catalase overexpression improves memory and reduces anxiety symptoms even in the absence of altered oxidative stress, and antidepressant treatment appears to increase levels of this antioxidant enzyme in patients with depression [29,32]. Correspondingly, the same occurs concerning physical exercise; Sousa et al. [33] demonstrated in a meta-analysis that physical exercise seems to promote increased antioxidant defense. Similarly, our study showed an increased activity in antioxidant defenses, mainly catalase, and a negative correlation between the intensity of depression symptoms and catalase activity in master athletes.

### Limitations

Possible limitations of this study may include that we did not measure inflammatory indicators. However, the correlation between depression and inflammation is already well described in the scientific literature. Despite this, we studied a group of high-level master athletes with a track record of long-term sprint and endurance training and success in national and/or international championships. Thus, to the best of our knowledge, this is the first study evaluating and comparing the intensity of depression symptoms and antioxidant parameters in elite master athletes, middle-aged, and young individuals with no lifelong training history.

## 5. Conclusions

In conclusion, master sprinters presented the lowest intensity of depression symptoms, with CAT being higher than CO and ER. CAT and the CAT/TBARS ratio were negatively associated with the intensity of depression symptoms, suggesting that the training model of master sprinters may be more effective in increasing CAT and reducing depressive symptoms. As a general recommendation, the lifestyle of master athletes, which is mainly characterized by, but not limited to, a lifetime of exercise training, seems to promote a better antioxidant defense system, favoring the redox balance. A better antioxidant defense system is related to a lower intensity of depressive symptoms and attenuates the aging process, as documented in several previous studies [3,4,32]. Thus, individuals seeking these benefits should exercise chronically at proper doses according to their preferences, in addition to maintaining a lifestyle with other healthy habits such as a balanced diet, proper sleep, and stress management.

## Figures and Tables

**Figure 1 ijerph-20-04397-f001:**
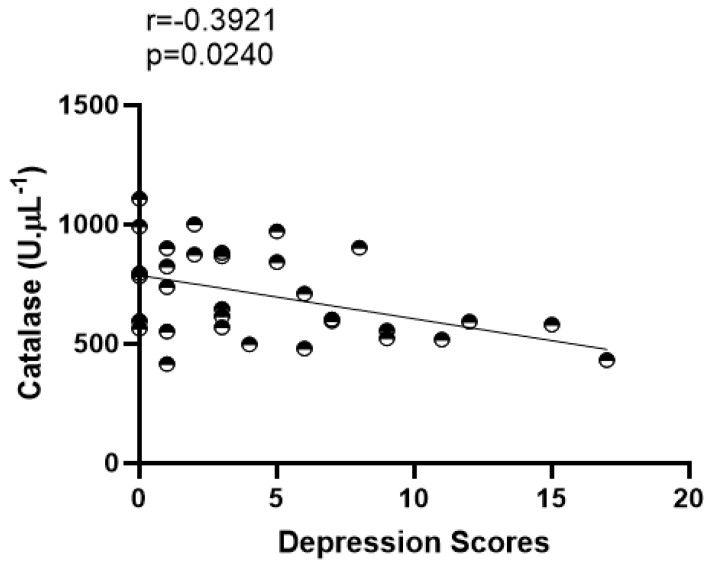
Correlation analysis among catalase and intensity of depression symptoms in master athletes.

**Figure 2 ijerph-20-04397-f002:**
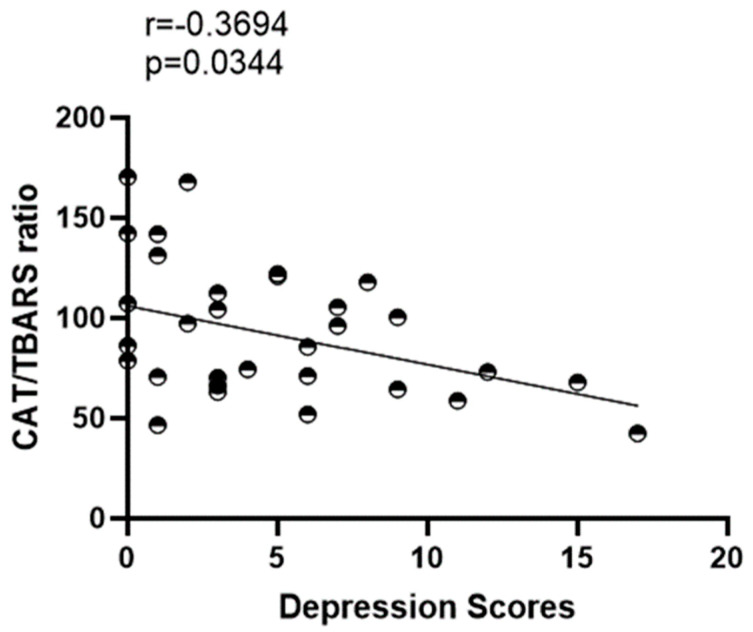
Correlation analysis among CAT/TBARS ratio and intensity of depression symptoms in master athletes.

**Figure 3 ijerph-20-04397-f003:**
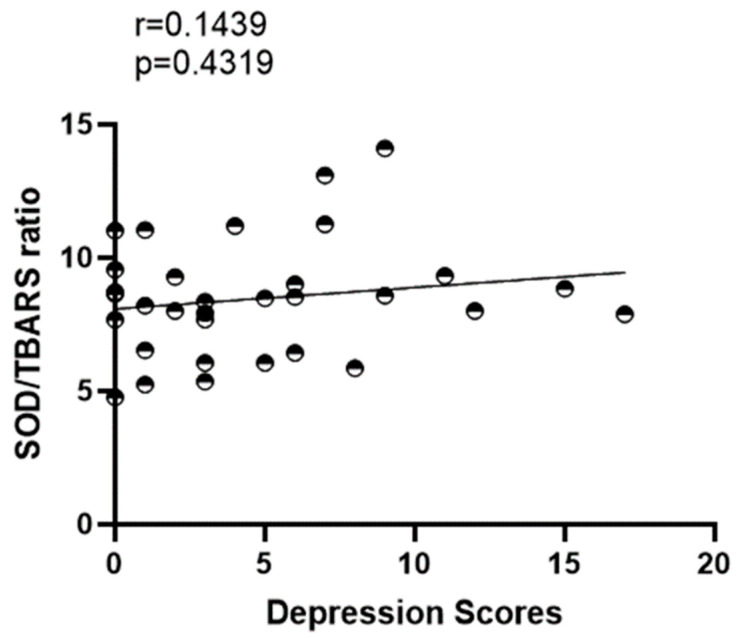
A correlation analysis was conducted among the ratio of SOD/TBARS and the intensity of depression symptoms in master athletes.

**Table 1 ijerph-20-04397-t001:** Age, intensity of depression symptoms, catalase, SOD, and Tbars of UY, CO, MS, and ER.

Variables	UY	CO	MS	ER	*p*-Value
Age (years)					
Mean and SD.	23.70 ± 4.02 ^b,c,d^	47.21 ± 8.61	50.31 ± 6.34	51.35 ± 9.12	<0.0001
Minimum	19.00	35.00	38.00	35.00	
25% Percentile	21.00	42.00	45.00	42.00	
Median	22.00	46.00	51.00	52.50	
75% Percentile	27.00	50.00	55.50	58.00	
Maximum	33.00	65.00	62.00	66.00	
The intensity of depression symptoms					
Mean and SD.	12.27 ± 9.27 ^b,c^	4.61 ± 2.56	3.60 ± 3.66	7.66 ± 5.29	0.0002
Minimum	1.000	0.000	0.000	1.000	
25% Percentile	7.000	3.500	0.5000	2.500	
Median	10.00	5.000	3.000	8.000	
75% Percentile	16.00	6.500	6.000	11.50	
Maximum	39.00	8.00	15.00	17.00	
Catalase (U·μL^−1^)					
Mean and SD.	729.9 ± 186.9 ^b,d^	410.3 ± 67.24	760.4 ± 170.1 ^b,d^	528.8 ± 103.2	<0.0001
Minimum	318.8	301.1	523.5	415.0	
25% Percentile	592.5	347.1	595.5	443.8	
Median	801.1	420.6	783.5	508.0	
75% Percentile	882.7	467.4	891.4	584.2	
Maximum	956.6	494.4	1108	738.2	
SOD (U·mL^−1^)					
Mean and SD.	84.20 ± 8.52 ^b,c^	53.26 ± 13.05	61.46 ± 11.94	78.24 ± 6.59 ^b,c^	<0.0001
Minimum	71.11	32.14	40.95	65.12	
25% Percentile	76.12	43.90	48.71	74.36	
Median	85.83	54.34	65.38	80.28	
75% Percentile	92.05	61.88	71.27	82.94	
Maximum	95.92	73.22	75.64	85.82	
Tbars (nmol·L^−1^)					
Mean and SD.	6.29 ± 1.47 ^d^	11.97 ± 2.35 ^a,c,d^	7.63 ± 1.39	8.88 ± 1.17	<0.0001
Minimum	4.170	7.990	5.210	6.700	
25% Percentile	4.630	9.860	6.310	8.258	
Median	6.380	11.63	7.810	8.860	
75% Percentile	7.990	14.33	8.790	9.930	
Maximum	8.240	14.94	9.990	10.46	

A one-way ANOVA followed by Tukey′s post hoc was applied to compare the groups. UY: young individuals; CO: control older; MS: sprint master athletes; ER: endurance runners in age, catalase, and Tbars variables. Kruskal-Wallis with Dunn’s test of multiple comparisons was applied to compare the groups. UY: young individuals; CO: control older; MS: sprint master athletes; ER: endurance running on depression and SOD variables. (a) *p* < 0.05 in comparison to the young individuals. (b) *p* < 0.05 in comparison to the control older. (c) *p* < 0.05 in comparison to the sprint master athletes group. (d) *p* < 0.05 in comparison to the endurance runners. A significance level of *p* < 0.05 was adopted.

## Data Availability

The data that support the findings of this study are available from the corresponding author upon reasonable request.

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
