# Peer review of "Intensity of Depression Symptoms Is Negatively Associated with Catalase Activity in Master Athletes"

_ijerph, 2023, doi:10.3390/ijerph20054397_

Round 1
Reviewer 1 Report
Dear Authors, in my opinion the article is interesting and it could be published in IJERPH after making some minor revisions regarding the explanation and correction of the issues presented below:
It is not clear why only catalase and superoxide dismutase activities and TBARS plasma concentration have been selected as markers of prooxidant-antioxidant balance.
It is not clear why the correlation between SOD activity as well TBARS concentration and the intensity of depression symptoms have not been statistically verified.
Table 1 is not properly designed, as in actual version it is not clear if the p-values in 6th (right) column refer to the comparison of four groups together. If yes, the statistical verification of the comparison between particular groups is lacking. It is not clear what symbols a, b, c and d presented as superscripts mean. This should be precisely clarified in the description below the table 1.
In Discussion chapter the difference between the results obtained in MS and ER groups (including the athletes in similar age and training experience) have not been sufficiently discussed and explained, though those are in my opinion also very interesting findings of the performed study.
Except of lacking assessment of inflammatory markers other limitation of the study is selection of only two antioxidant enzymes and only one marker of oxidative stress - that makes the interpretation of the impact of training on parameters of prooxidant and antioxidant balance very difficult or even impossible.
The references list is not acceptable and it must be extensively corrected, as in case of many references numerous bibliographic data are lacking (journal names, years of publication, numbers of volumes or issues, numbers of pages and even parts of article titles). Moreover the references list should be presented in uniform style of IJERPH.
Author Response
Dear reviewer:
We welcome comments and suggestions on the manuscript. As a result, the manuscript was revised and we are resubmitting it.
Below are the responses to the comments.
RESPONSES TO REVIEWER 1
Comment: Dear Authors, in my opinion the article is interesting and it could be published in IJERPH after making some minor revisions regarding the explanation and correction of the issues presented below:
It is not clear why only catalase and superoxide dismutase activities and TBARS plasma concentration have been selected as markers of prooxidant-antioxidant balance.
Response: Dear Reviewer, We exclusively chose catalase and superoxide dismutase because they are primary antioxidant enzymes. In addition, the differences between sprint and endurance training outcomes may result in disparate adaptations in the antioxidant defense system. For example, endurance-trained athletes are more prone to mitochondrial biogenesis, while sprint trained ones may be better adapted on the glycolytic system (cytosolic). Thus, these specialists stress their body and metabolism specifically through different metabolic pathways, which in turn may influence the expression rate of enzymes such as superoxide dismutase (more mitochondrial) and catalase (more cytosolic). We chose TBARS because we could observe the redox balance of each group of athletes by calculating the superoxide dismutase/TBARS and catalase /TBARS ratios.
It is not clear why the correlation between SOD activity as well TBARS concentration and the intensity of depression symptoms have not been statistically verified.
These statistics were verified, but we did not find any significant correlations between SOD, TBARS, and the intensity of depression symptoms. However, because of your comment, we inserted the following sentence in the results section:
“The relationships between the intensity of depression symptoms with SOD [r=0.3320 (p=0.06)] and TBARS [r=0.0900 (p=0.61)] were not statistically significant.”
Table 1 is not properly designed, as in actual version it is not clear if the p-values in 6th (right) column refer to the comparison of four groups together. If yes, the statistical verification of the comparison between particular groups is lacking. It is not clear what symbols a, b, c and d presented as superscripts mean. This should be precisely clarified in the description below the table 1.
Thank you for the observation. The p values in the 6th (right) column refer to the comparison among all 4 groups together, and the values for the comparisons between specific groups are now described in the legend below the table, as well as the meaning of each superscripted symbol.
“ap< 0.05 in comparison to the young individuals. bp< 0.05 in comparison to the control older. cp< 0.05 in comparison to the sprint master athletes group. dp< 0.05 in comparison to the endurance runners.”
In Discussion chapter the difference between the results obtained in MS and ER groups (including the athletes in similar age and training experience) have not been sufficiently discussed and explained, though those are in my opinion also very interesting findings of the performed study.
As with CAT, several studies have demonstrated the role of physical exercise in increasing SOD activity. In the present study, the UY and ER groups showed higher SOD activity when compared to the other groups. The differences between sprint and endurance training outcomes may result in disparate adaptations in the antioxidant defense system. For example, endurance-trained athletes are more prone to mitochondrial biogenesis, while sprint trained ones may be better adapted on the glycolytic system, thus influencing the expression of enzymes such as SOD, which has mitochondrial predominance. Despite the differences found between groups on SOD activity, the intensity of depression symptoms was not correlated with SOD. Furthermore, high levels of NOx were found in master endurance athletes, which in turn promote an inhibitory effect on CAT activity. This fact could also explain these differences between the two types of training modalities, as described in present study. Some previous studies from our research group support our hypothesis.
Aguiar, S.S.; Sousa, C.V.; Deus, L.A.; Rosa, T.S.; Sales, M.M.; Neves, R.V.P.; Barbosa, L.P.; Santos, P.A.; Campbell, C.S.; Simões, H.G.J.E.g. Oxidative stress, inflammatory cytokines and body composition of master athletes: the interplay. 2020, 130, 110806.
Rosa, T.S.; Neves, R.V.P.; Deus, L.A.; Sousa, C.V.; da Silva Aguiar, S.; de Souza, M.K.; Moraes, M.R.; Rosa, É.C.C.C.; Andrade, R.V.; Korhonen, M.T.J.N.O. Sprint and endurance training in relation to redox balance, inflammatory status and biomarkers of aging in master athletes. 2020, 102, 42-51.
Except of lacking assessment of inflammatory markers other limitation of the study is selection of only two antioxidant enzymes and only one marker of oxidative stress - that makes the interpretation of the impact of training on parameters of prooxidant and antioxidant balance very difficult or even impossible.
We agree that one of the limitations of this study was the choice of only two antioxidant enzymes and one marker of oxidative stress. However, this study was conducted in two stages. In the first phase of data collection, we published several articles with other antioxidant enzymes, oxidative stress, and inflammatory markers, showing the difference between them in different training models performed by master athletes. Unfortunately, at that time, we did not assess depression symptoms.
Aguiar, S.S.; Rosa, T.S.; Sousa, C.V.; Santos, P.A.; Barbosa, L.P.; Deus, L.A.; Rosa, E.C.; Andrade, R.V.; Simões, H.G.J.T.J.o.S.; Research, C. Influence of body fat on oxidative stress and telomere length of master athletes. 2021, 35, 1693-1699.
Aguiar, S.S.; Sousa, C.V.; Deus, L.A.; Rosa, T.S.; Sales, M.M.; Neves, R.V.P.; Barbosa, L.P.; Santos, P.A.; Campbell, C.S.; Simões, H.G.J.E.g. Oxidative stress, inflammatory cytokines and body composition of master athletes: the interplay. 2020, 130, 110806.
Rosa, T.S.; Neves, R.V.P.; Deus, L.A.; Sousa, C.V.; da Silva Aguiar, S.; de Souza, M.K.; Moraes, M.R.; Rosa, É.C.C.C.; Andrade, R.V.; Korhonen, M.T.J.N.O. Sprint and endurance training in relation to redox balance, inflammatory status and biomarkers of aging in master athletes. 2020, 102, 42-51.
The references list is not acceptable and it must be extensively corrected, as in case of many references numerous bibliographic data are lacking (journal names, years of publication, numbers of volumes or issues, numbers of pages and even parts of article titles). Moreover the references list should be presented in uniform style of IJERPH.
All references have been adjusted according to the style of IJERPH.
Reviewer 2 Report
1.- I suggest to include , information about the type of training, because I understand that they are master elite male athletes at a regional, national and international level with specialized physical training that combines resistance and resistance exercises. It is important because the physiological changes that occur during exercise, depend on the type of exercise (aerobic or anaerobic), intensity, duration etc.
2.- As they are elite ahtletes, does their training program include any psychological counseling? Sice they are elite athlets with a systematic training for at least 10 years, probably have the capacity to handle the symptoms of depression which could bias the results
3.- Why did you include a young control group with an age two times less than the trained groups? It is clear that there would be differences in depression degree between young and older people. I consider that it is not convenient to compare the elite athletes with the young control group.
4.- Do the young untrained group were profesionist, empoyes, unemployes, students, singles, married, divorced? Do they were recluted from Brazil, Uruguay and Poland? It is striking that this population has the highest intensity of depression symptoms. It would be convenient to include demographic data of your populations
5.-When did you apply the Beck Depression Inventory-II (BDI-II)? During recruitment, at the begining or at the end of the competitions?
6.- How many of the athletes won or lost in their competitions? This data may influence the score of Beck Inventory
7.- How many participants were classified with minimal, mild, moderate, and severe depression and what were the catalase levels by classification? These data should be included in the results.
8.- Define a,b,c,d of the table
Author Response
Dear reviewer:
We welcome comments and suggestions on the manuscript. As a result, the manuscript was revised and we are resubmitting it.
Below are the responses to the comments.
RESPONSES TO REVIEWER 2
1.- I suggest to include, information about the type of training, because I understand that they are master elite male athletes at a regional, national and international level with specialized physical training that combines resistance and resistance exercises. It is important because the physiological changes that occur during exercise, depend on the type of exercise (aerobic or anaerobic), intensity, duration etc.
Dear Reviewer, Thank you for your suggestion to help us improve the manuscript. As a result of your comments, the following sentence has been added in the Introduction:
“These master athletes undergo intense physical training routines (from 3 to 6 sessions per week, totaling approximately 10 hours or more of weekly training). The training programs of elite masters sprinters, for example, are mainly characterized by high-intensity, low-volume sessions, based mainly on anaerobic pathways. Additionally, weekly training of such sprinters usually includes two sessions for speed (i.e., short sprints and long sprints for speed endurance), strength (i.e., weight lifting), and power (i.e., plyometric exercises) and stretching and flexibility. A few low-volume and moderate-intensity cardiovascular sessions can also be performed between high-intensity anaerobic training sessions. Otherwise, the training programs of elite master endurance athletes are based mainly on lower-intensity, high-volume sessions, relying primarily on aerobic pathways. Exercise modes and workloads are selected individually depending on the athlete’s training season.”
Simoes, H. G.; Sousa, C. V.; Dos Santos Rosa, T.; Da Silva Aguiar, S. et al. Longer telomere length in elite master sprinters: relationship to performance and body composition. 38, n. 14, p. 1111-1116, 2017.
Korhonen, M. T.; Haverinen, M.; Degens, H. J. N.; Athletes, p. i. m. 16 training and nutritional needs. p. 291, 2014.
2.- As they are elite ahtletes, does their training program include any psychological counseling? Sice they are elite athlets with a systematic training for at least 10 years, probably have the capacity to handle the symptoms of depression which could bias the results.
We did not verify with the participants whether they had psychological counseling prior to their enrollment in the study. We agree that being an athlete seems to be important for having a lower intensity of depressive symptoms when compared to young untrained individuals.
3.- Why did you include a young control group with an age two times less than the trained groups? It is clear that there would be differences in depression degree between young and older people. I consider that it is not convenient to compare the elite athletes with the young control group.
Our initial hypothesis was that we would find differences between groups. However, our findings were not completely in agreement with our hypothesis. We expected to find higher levels of depressive symptoms in the control group of non-athlete middle-aged individuals, given that the literature shows an increase in the prevalence and incidence of depression as people age, and the same did not happen with master athletes. In order to better understand the impact of aging, we also included a young group as another control (for the age effect). However, what we found was that intensity of depressive symptoms was greater in the young group, which could be explained by greater emotional pressures in this phase of life and other environmental factors not addressed in our study.
4.- Do the young untrained group were profesionist, empoyes, unemployes, students, singles, married, divorced? Do they were recluted from Brazil, Uruguay and Poland? It is striking that this population has the highest intensity of depression symptoms. It would be convenient to include demographic data of your populations
The youth sample was entirely collected in Brazil. These were mostly single college students. We actually found interesting results compiled in a systematic review with a meta-analysis demonstrating that university students have a higher intensity of depressive symptoms, with singles having higher levels than married ones.
Sarokhani, Diana et al. Prevalence of depression among university students: a systematic review and meta-analysis study. Depression research and treatment, vol. 2013, 2013.
It is hypothesized that young adults are prone to anxiety and depression due to their greater life pressure and uncertainties, such as the possible lack of employment and the fear to face financial problems.
5.-When did you apply the Beck Depression Inventory-II (BDI-II)? During recruitment, at the begining or at the end of the competitions?
The questionnaires were administered one day before the competitions at the athlete’s accommodation, which was usually close to the competition venue. The same researcher performed all of these assessments.
6.- How many of the athletes won or lost in their competitions? This data may influence the score of Beck Inventory
This is an interesting question, but since we collected the depression symptoms prior to the competition, we would not be able to answer it. We also did not score the BDI immediately after data collection, so the participant did not know his results. Thus, this assessment should not have interfered with the athlete’s participation and ultimately on winning or losing the competition. This topic could be addressed in future studies by assessing the symptoms before and after the competition. We speculate that those with lower scores of both depression and anxiety would be likely to better control emotions and behaviors to benefit performance.
7.- How many participants were classified with minimal, mild, moderate, and severe depression and what were the catalase levels by classification? These data should be included in the results.
We did not divide the sample by classification range due to lack of variance in the values. In our control older sample, all participants scored at the minimal level, and in the groups of athletes only one participant from each group scored at a mild level, while all the others were at the minimal level. In the group of young individuals, only one volunteer scored at a severe level and one scored at a minimal level. Thus, we do not have a large enough sample of participants to stratify by the levels of depression symptom severity.
8.- Define a,b,c,d of the table
As noted above for Reviewer 1, we have the added values for the comparisons between specific groups that are now described in the legend below the table, as well as the meaning of each superscripted symbol.
“ap< 0.05 in comparison to the young individuals. bp< 0.05 in comparison to the control older. cp< 0.05 in comparison to the sprint master athletes group. dp< 0.05 in comparison to the endurance runners.”
Reviewer 3 Report
His study examined associations between scores of depression (DEPs), thiobarbituric acid reactive substances (TBARS), superoxide dismutase (SOD), and catalase activity (CAT) in master 19 athletes and untrained controls.
The work seems interesting but I have a few doubts:
Abstract: Divide it into sections : Background, methods, results, conclusion
Introduction: As an introduction, a broader literature review should be conducted
Material and methods:
Material and method are well written, information on sample size determination is missing. In addition, ranges of correlation should be noted in the description of the statistical analysis. Additionally, why was Sperman's rank correlation used? in the abstract you mention persona correlation however I do not see this in the description of the statistical analysis?
Results:
Change the notation in the table from P to p
Where you specified effect size ?
Discussion
limitations in a separate subchapter
Conclusion
Does your study have any practical application?
Author Response
Dear reviewer:
We welcome comments and suggestions on the manuscript. As a result, the manuscript was revised and we are resubmitting it.
Below are the responses to the comments.
RESPONSES TO REVIEWER 3
His study examined associations between scores of depression (DEPs), thiobarbituric acid reactive substances (TBARS), superoxide dismutase (SOD), and catalase activity (CAT) in master 19 athletes and untrained controls.
The work seems interesting but I have a few doubts:
Abstract: Divide it into sections: Background, methods, results, conclusion
Dear reviewer, The Abstract has been divided into sections as you suggested.
Introduction: As an introduction, a broader literature review should be conducted
We have added additional detail as per your suggestion:
The antioxidant defense system, in turn, protects cells by removing free radicals. This antioxidant system comprises different types of functional components, such as superoxide dismutase (SOD) and catalase (CAT). SOD acts as a primary cellular defense against free radicals, since it catalyzes the reduction of SO to oxygen and hydrogen peroxide. CAT is an antioxidant enzyme present in almost all aerobic organisms. Its function is to break two molecules of hydrogen peroxide into one molecule of oxygen and two molecules of water.
Deisseroth, A.; Dounce, A.L.J.P.r. Catalase: Physical and chemical properties, mechanism of catalysis, and physiological role. 1970, 50, 319-375.
Material and methods:
Material and method are well written, information on sample size determination is missing. In addition, ranges of correlation should be noted in the description of the statistical analysis. Additionally, why was Sperman's rank correlation used? in the abstract you mention persona correlation however I do not see this in the description of the statistical analysis?
The following sentence has been added in the Methods:
“The sample size for a priori statistical power of 80% (1-β=0.80) indicated 20 participants for a significance level of 5% (α=0.05) and small effect size (f=0.4). Thus, we chose a sample of 80 subjects (20 for each studied group).”
The data were analyzed according to their normality. If the data were normal, they were analyzed with Pearson’s correlation. If the data were not normal, they were analyzed with Spearman’s correlation.
FIELD, Andy. Discovering statistics using IBM SPSS statistics. sage, 2013.
Results:
Change the notation in the table from P to p
The change has been made as per your guidance.
Where you specified effect size?
The effect size was included in the study methods.
Discussion
Limitations in a separate subchapter
We have added a subsection for Limitations.
Conclusion
Does your study have any practical application?
Yes, it does. Thank you for this comment. As a result, the following sentence has been added in the Conclusion section:
“As a general recommendation, the lifestyle of master athletes, which is mainly characterized by, but not limited to, a lifetime of exercise training, seems to promote a better antioxidant defense system, favoring the redox balance. A better antioxidant defense system is related to a lower intensity of depressive symptoms and attenuates the aging process as documented in several previous studies [3, 4, 32]. Thus, individuals seeking these benefits should exercise chronically at proper doses according to their preferences, in addition to maintaining a lifestyle with other healthy habits, such as balanced diet, proper sleep, and stress management.”
- Aguiar, S.S.; Sousa, C.V.; Deus, L.A.; Rosa, T.S.; Sales, M.M.; Neves, R.V.P.; Barbosa, L.P.; Santos, P.A.; Campbell, C.S.; Simões, H.G.J.E.g. Oxidative stress, inflammatory cytokines and body composition of master athletes: the interplay. 2020, 130, 110806.
- Rosa, T.S.; Neves, R.V.P.; Deus, L.A.; Sousa, C.V.; da Silva Aguiar, S.; de Souza, M.K.; Moraes, M.R.; Rosa, É.C.C.C.; Andrade, R.V.; Korhonen, M.T.J.N.O. Sprint and endurance training in relation to redox balance, inflammatory status and biomarkers of aging in master athletes. 2020, 102, 42-51.
- de Sousa, C.V.; Sales, M.M.; Rosa, T.S.; Lewis, J.E.; de Andrade, R.V.; Simões, H.G.J.S.m. The antioxidant effect of exercise: a systematic review and meta-analysis. 2017, 47, 277-293.
Round 2
Reviewer 2 Report
1.- I consider that this pharagraph should be included in the General procedures section, subsection b) "The questionnaires were administered one day before the competitions at the athlete’s accommodation, which was usually close to the competition venue. The same researcher performed all of these assessments".
2.- Also include this information in 2.2 Participants section: The youth sample was entirely collected in Brazil. These were mostly single college students.
Author Response
Dear reviewer:
We welcome comments and suggestions on the manuscript. As a result, the manuscript was revised and we are resubmitting it.
Below are the responses to the comments.
1.- I consider that this pharagraph should be included in the General procedures section, subsection b) "The questionnaires were administered one day before the competitions at the athlete’s accommodation, which was usually close to the competition venue. The same researcher performed all of these assessments".
Dear Reviewer, Thank you for your suggestion to help us improve the manuscript. As a result of your comments, the following sentence has been added in the general procedures section, subsection b):
"The questionnaires were administered one day before the competitions at the athlete’s accommodation, which was usually close to the competition venue. The same researcher performed all of these assessments".
2.- Also include this information in 2.2 Participants section: The youth sample was entirely collected in Brazil. These were mostly single college students.
Information has been included as suggested.
Reviewer 3 Report
The authors have revised the paper as suggested. He therefore suggests its publication in its current form
Author Response
Dear reviewer, thank you for your comment.